# Building 3D Representations and Generating Motions From a Single Image via Video-Generation

**Weiming Zhi** [1,2,3]      **Ziyong Ma** [3]      **Tianyi Zhang** [3,4]

**Matthew Johnson-Roberson** [2,3]

[1] School of Computer Science, The University of Sydney, Australia.
[2] College of Connected Computing, Vanderbilt University, TN, USA.
[3] Robotics Institute, Carnegie Mellon University, PA, USA.
[4] Aurora, USA.
Correspondence to W. Zhi (`Weiming.zhi@sydney.edu.au`).

## Abstract

Autonomous robots typically need to construct representations of their surroundings and adapt their motions to the geometry of their environment. Here, we tackle the problem of constructing a policy model for collision-free motion generation, consistent with the environment, from a single input RGB image. Extracting 3D structures from a single image often involves monocular depth estimation. Developments in depth estimation have given rise to large pre-trained models such as *DepthAnything*. However, using outputs of these models for downstream motion generation is challenging due to frustum-shaped errors that arise. Instead, we propose a framework known as Video-Generation Environment Representation (VGER), which leverages the advances of large-scale video generation models to generate a moving camera video conditioned on the input image. Frames of this video, which form a multiview dataset, are then input into a pre-trained 3D foundation model to produce a dense point cloud. We then introduce a multi-scale noise approach to train an implicit representation of the environment structure and build a motion generation model that complies with the geometry of the representation. We extensively evaluate VGER over a diverse set of indoor and outdoor environments. We demonstrate its ability to produce smooth motions that account for the captured geometry of a scene, all from a single RGB input image.

## 1 Introduction

Autonomous robots operating in unstructured environments maintain a representation of their surroundings to enable reliable motion planning. Building complex environment representations generally requires large sets of images [1–3], and often depth measurements [4, 5]. In this work, we push the boundaries of data efficiency by constructing both a scene representation for collision avoidance and a downstream motion generation model from a single RGB image, by leveraging recent advances in pre-trained video generation and 3D foundation models.

A common strategy for recovering 3D structure from a single image is to estimate per-pixel depth. Recent advances have yielded powerful monocular depth estimators, most notably *DepthAnything*. However, the depth maps they produce often exhibit frustum-shaped artifacts at object boundaries, rendering them unsuitable for collision avoidance. To overcome these limitations, we introduce Video-generation Environment Representation (VGER), which sidesteps direct monocular depth prediction by conditioning a large, pre-trained video model on the input image to generate a se-

39th Conference on Neural Information Processing Systems (NeurIPS 2025).

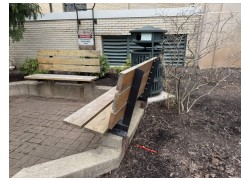 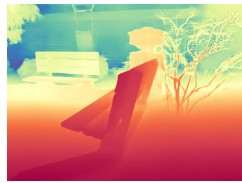 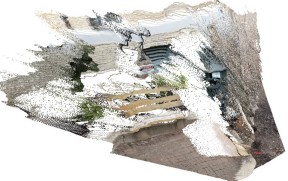 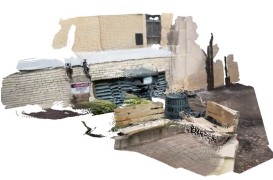

|(a) Input image | (b) DepthAnything-V2 image (l) and structure (r) | (c) VGER |

Figure 1: (a): Single original image used to construct the environment; (b): Predictions by DepthAnything-V2 [9]. Predicted depth image of the left, and extracted structure on the right. We observed errors that are frustum-shaped "tails" in the extracted structure; (c): Our proposed VGER does not suffer from these errors, and completes regions blocked from the original view. These artifacts in free space make it impossible to use the representation for motion generation.

quence of consistent novel views. We then apply a 3D foundation model such as DUSt3R [6] to fuse these views into a dense representation that faithfully captures scene geometry.

Beyond representation, our primary contribution lies in coupling the structure extracted from the single input image with a structured motion-generation framework. We integrate our reconstruction into motion generation by deriving an implicit unsigned distance field from the 3D foundation model's output. We do so efficiently via multi-scale perturbations. By sampling perturbations of point positions at varying noise scales and contrasting them against the dense output, we recover an implicit, unsigned distance field that defines a continuous distance function over the workspace. From this, we construct a smoothly varying metric field that encodes proximity to obstacles. Finally, we embed this environment-dependent metric field into a motion policy framework, in a similar manner to Riemannian Motion Policies [7] and Diffeomorphic Templates [8], in which the environment-induced metric modulates a nominal dynamical system. The result is a policy that generates motion and is modelled by a non-linear dynamical system that adapts fluidly to the reconstructed geometry, producing smooth, collision-free trajectories directly from a single input image.

Concretely, our technical contributions within VGER include,

1. A pipeline that harnesses pretrained video synthesis and 3D foundation models to generate multi-view videos from a single RGB image, producing dense, artifact-free reconstructions that overcome the limitations of monocular depth estimators;

2. A multi-scale noise-contrastive denoising approach to extract a globally detailed implicit unsigned distance field directly from the generated output;

3. A novel metric-modulated motion generation framework that embeds the implicit representation into a metric field that is dependent on the environment. This ensures that any nominal dynamical system coupled with the obstacle-induced curvature metric field will yield smooth, collision-free trajectories in real time.

## 2 Related Work

**Pre-trained Models to Recover 3D Structure RGB Images:** Estimating the structure within a single RGB image typically requires the estimation of depth, i.e. monocular depth estimation. Advances in deep learning have lead to large pre-trained models capable of predicting a depth image given an RGB input. Well-known models include Marigold [10], DepthAnything [11], and its follow-up model DepthAnything-V2 [9]. However, extracting structure from learned depth estimates can often lead to artifacts. Our proposed VGER differs in that it seeks to generate an image sequence from the single input image, and extract the structure that jointly appears.

Recovering 3D structure from a set of images is another well-known problem known as *structure-from-motion* [12]. Classical methods, such as COLMAP [13], produce relatively sparse representations. Modern advances in deep learning have led to transformer models capable of efficiently constructing highly dense 3D representations in a single feed-forward pass, including DUSt3R [6], MASt3R [14] and Visual Geometry Grounded Transformers (VGGT) [15]. These models typically act as *3D foundation models*, which are large pre-trained models, intended as plug-and-play modules

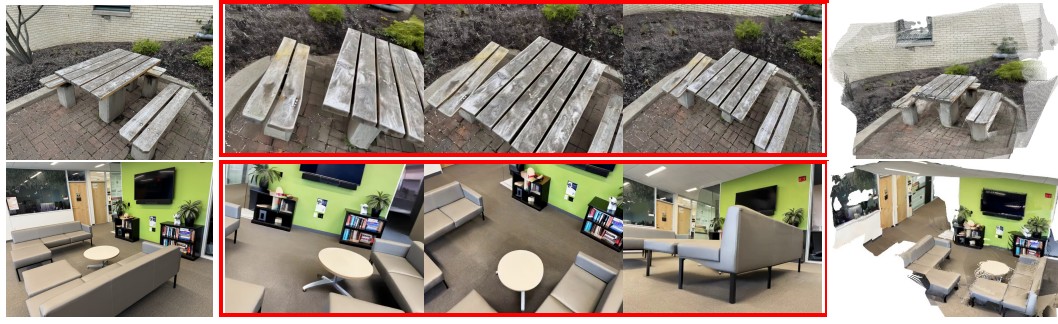

| (a) Input Image | (b) Frames of generated input-conditioned video | (c) 3D structure |

Figure 3: Examples of extracting a 3D structure of an outdoor bench (top) and indoor office environment (bottom) from input images. Input images are shown in subfigure (a). We leverage a video generator, conditional on the images, to generate videos, with frames shown in subfigure (b). These are then subsequently used to construct 3D structures via foundation models, without any frustum-shaped artifacts, shown in subfigure (c).

for downstream tasks. These 3D foundation models have found use in a variety of robot perception and calibration problems [16].

**Video Generation Models:** A recent wave of deep learning–based general video generators spans unconditional, text-to-video, and image-to-video synthesis. Early GAN-based methods (StyleGAN-V [17]) gave way to diffusion and transformer systems: Video Diffusion Models [18] produce high-quality clips, and MagicVideo [19] extends diffusion priors for text-driven generation. Many recent advances in this field have focused on the controllability of the camera motions in the video generator, these include GEN3C [20] and ViewCrafter[21], and SEVA [22]. Particularly, in this work, we leverage a recent video generator Stable Virtual Camera (Seva) [22], which can condition on an image as well as a camera trajectory to generate a video that follows the trajectory in the scene of the image.

**Reactive Motion Generation:** Our work is the first to apply structures and representations produced by 3D foundation models for reactive motion generation. Relevant frameworks, including Riemannian Motion Policies [7] and Geometric Fabrics [23, 24], to generate motion creatively generally approach the generation from a planning perspective and require the assumption of a pre-defined surface, or a completely known environment constructed from simple shape primitives. Other robotics focused representations may require depth [25] or velocity measurements [26]. There has been some effort to extend from representing environments to meshes instead of pre-designed primitives [27]. However, there is a clear disconnect between the literature on scene representation and motion generation. To bridge this gap, Our method is unique in extracting both a representation of the structure and a subsequent motion generation model from a single RGB image.

## 3 Video generation Environment Representation: Structure Extraction to Motion Generation From a Single Image

Here, we elaborate on the technical details of the VGER method. This includes: leveraging pre-trained video generators to extract a structure conditional on the input image (section 3.1); the construction of an implicit model via multi-scale noise contrastive samples (section 3.2); generating motion from an environment-dependent metric field (section 3.3), giving collision-free motion. An overview of the workflow is shown in fig. 2.

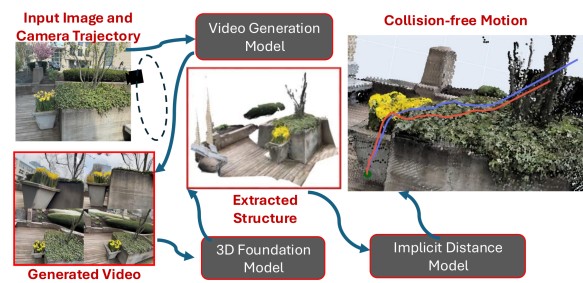

Figure 2: Pipeline of VGER.

## 3.1 Structure Extraction from Single Image with Pre-trained Video Generators

We circumvent the direct prediction of any depth information from the input image, thereby avoiding frustum-shaped artifacts that are characteristic. Instead, we leverage modern video generators to synthesize a sequence of images of the camera moving, conditional on the input image. These images are then passed into a 3D foundation model, such as DUSt3R [6], to produce a dense point cloud. We use the pre-trained video generation model, Stable Virtual Camera (Seva) [22]. Seva is a 1.3 billion parameter diffusion model, using a stable diffusion 2.1 backbone [28]. It produces a sequence of output images $\{I_1, I_2, \ldots, I_n\}$, conditional on the input image $I_0$ and a trajectory of camera poses $\{T_1, \ldots, T_n\}$, where $n$ is a pre-determined image sequence length. Despite the video generator not constructing an explicit 3D representation, Seva is trained on videos that maintain both photometric and geometric consistency across frames. As a result, the generated image sequences maintain geometry and temporal consistency.

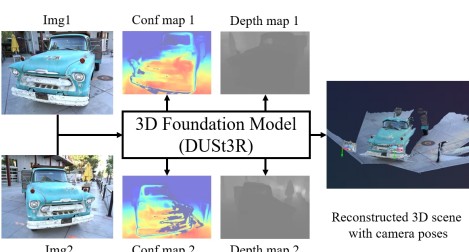

Figure 4: We leverage DUSt3R [6], which can produce 3D structures from sets of 2D images, and filter based on confidence maps.

The generated images, along with the conditioned input image, are then all inputted into the 3D foundation model, DUSt3R [6]. DUSt3R is a fully data-driven system built around a large vision transformer [29]. It produces dense 3D point-maps along with per-pixel confidence and depth estimates, and then uses these to recover relative camera poses and a fully consistent 3D reconstruction. Because it learns correspondences end-to-end rather than relying on hand-crafted features, it can accurately estimate pose even when the two views have very little visual overlap. After filtering away the points with low confidence, we recover a dataset of dense cloud of $N$ 3D points $\{p_i, c_i\}_{i=1}^{N}$, where $p_i \in \mathbb{R}^3$ gives the 3D coordinates and $c_i$ gives the RGB colors of the point. Next, we seek to construct a continuous implicit representation of the environment from the point cloud. This implicit model enables smooth geometry reconstruction and supports downstream tasks such as collision avoidance and motion generation.

## 3.2 Multi-scale Noise Contrastive Implicit Model

Reactive motions generated in the environment are influenced by distances to the nearest surface in the environment. Here, VGER converts the point-based output from the 3D foundation model into a smoothly varying function that captures the distance to the nearest surface in the scene. This implicit function is modelled by a neural network $f_\theta : \mathbb{R}^3 \to \mathbb{R}$ that approximates the unsigned distance to a surface captured in our point cloud $\mathcal{P} = \{p_i\}_{i=1}^{N}$. Our training procedure optimizes a loss:

$$\mathcal{L} = \mathcal{L}_{\text{fit}} + \alpha_{\text{surf}}\mathcal{L}_{\text{surf}} + \alpha_{\text{eik}}\mathcal{L}_{\text{eik}}. \tag{1}$$

This loss combines: (i) regression to ground-truth distances at query points, (ii) a surface consistency term enforcing a zero level-set at the surface, and (iii) an Eikonal regularizer to encourage unit gradient norm. Linear weightings are denoted as $\alpha_{\text{surf}}$ and $\alpha_{\text{eik}}$. Next, we elaborate on this construct of each loss term.

**Multi-Scale Noise Negative Querying:** To train a model to recover distances both on and near the surface, we generate negative samples by perturbing points drawn from the surface with Gaussian noise at multiple scales. Concretely, at each iteration we first sample a batch of raw surface points $\{p_i\}_{i=1}^{N}$ uniformly from $\mathcal{P}$. Around each $p_i$, we add a zero-mean Gaussian noise $\varepsilon_j$ whose standard deviation $\sigma_j$ is itself drawn from a log-uniform distribution:

$$u_j \sim \text{Uniform}\big(\ln(\sigma_{\min}), \ln(\sigma_{\max})\big), \quad \sigma_j = \exp(u_j), \quad \varepsilon_j \sim \mathcal{N}(0, \sigma_j^2 I), \quad \hat{p}_{i,j} = p_i + \varepsilon_j.$$

Here, $\hat{p}_{i,j}$ denotes a *query point* that is perturbed off the surface point $p_i$ with a noise $\varepsilon_j$, and $\sigma_{\min}$ and $\sigma_{\max}$ give boundaries of the standard deviation of the noise. Sampling $\sigma_j$ on a log scale ensures that the network sees perturbations ranging from extremely fine to coarse (one-tenth of the bounding radius) distances. This multi-scale scheme serves two purposes: *Local surface fidelity:* Small $\sigma_j$ focuses the loss on points extremely close to the true surface, improving pointwise accuracy and helping the zero level set align precisely with $\mathcal{P}$; *Robust gradient learning:* Larger $\sigma_j$ forces the model to estimate distances further from the surface, ensuring that the predicted gradient field $\nabla f_\theta$

remains informative and prevents collapse to trivial solutions. Next, we compute the true unsigned distance efficiently via a KD-tree, and construct a mean squared error loss to learn a continuous distance representation:

$$d(\hat{p}) = \min_{p \in \mathcal{P}} \|\hat{p} - p\|, \qquad \mathcal{L}_{\text{fit}} = MSE(f_\theta(\hat{p}_{i,j}), d(\hat{p}_{i,j})).$$

The implicit model $f_\theta$ is represented by a neural network, that incorporates SIREN activation layers [30]. That is, each hidden layer applies a periodic activation:

$$h^{(l)} = \sin\big(\omega_0\,(W^{(l)}\,h^{(l-1)} + b^{(l)})\big), \quad l = 1, \dots, L.$$

Here, $h^{(l-1)}$ is the input to layer $l$, $W^{(l)} \in \mathbb{R}^{n_l \times n_{l-1}}$ and $b^{(l)} \in \mathbb{R}^{n_l}$ are the learned weights and biases, and $\omega_0 > 0$ is a fixed frequency scale controlling the periodicity. By fitting parameters in the frequency domain, these activation layers enable $f_\theta$ to capture finer details with higher frequency.

**Surface Consistency and Eikonal Regularization:** To ensure that our representation has *a priori* structure of $f_\theta(p_i) = 0$ and $\|\nabla f_\theta(x_i)\| = 1$, we provide additional regularization. For each training iteration, let $\mathcal{B}_s$ denote the minibatch of surface points and $\mathcal{B}_q$ the full minibatch of query points. To align the network's zero level-set with the input surface, and to encourage $\|\nabla f_\theta(x)\| = 1$ almost everywhere. We add a surface loss $\mathcal{L}_{\text{surf}}$ and a Eikonal loss $\mathcal{L}_{\text{eik}}$, given by

$$\mathcal{L}_{\text{surf}} = \frac{1}{|\mathcal{B}_s|} \sum_{p \in \mathcal{B}_s} |f_\theta(p)|, \qquad \mathcal{L}_{\text{eik}} = \frac{1}{|\mathcal{B}_q|} \sum_{\hat{p} \in \mathcal{B}_q} \big(\|\nabla_x f_\theta(\hat{p})\|_2 - 1\big)^2. \tag{2}$$

Together, these terms ensure exact surface interpolation and a well-conditioned gradient field, which promotes accurate normal estimation and stable mesh extraction.

**Connection to Diffusion Models:** Multi-scale denoising has proven crucial for sharpening implicit energy-based representations by teaching models to recover fine details at varying noise levels. In particular, frameworks such as denoising score matching [31] and score-based diffusion models [32] denoise data corrupted at multiple noise scales to learn a noise-conditional score

$$s_\theta(x, \sigma) \approx \nabla_x \log p_\sigma(x), \tag{3}$$

where $p_\sigma$ is the data distribution corrupted by Gaussian noise of scale $\sigma$. By sampling $\sigma$ log-uniformly over several orders of magnitude, these methods capture both high-frequency surface details (small $\sigma$) and global structure (large $\sigma$). Analogously, the multi-scale noise in VGER trains the implicit distance field $f_\theta$ to denoise points at various distances from the surface, effectively learning a landscape whose gradient aligns with the true distance gradient. This mirrors the reverse diffusion process, where successive denoising steps refine a sample towards the data manifold. Our incorporation of an Eikonal term $\mathcal{L}_{eik}$ further ensures that $\|\nabla f_\theta\| = 1$, enforcing the learned field into a distance metric and yielding sharp, artifact-free reconstructions.

## 3.3 Motion Generation in Constructed Environment via Continuous Metric Fields

Generating motion is central to enabling robots to interact with their surroundings. Here we outline how we can leverage the environment representations extracted from a single image to shape robot motions. We model robot policies as a dynamical system that is influenced by an environment-dependent metric field.

**Robot Policies as a Dynamical System:** We take a reactive dynamical systems approach, enabling motion at arbitrary initial positions in the environment. Similar to many of the reactive methods introduced in [7, 8], we model the task-space motion of a robot as an autonomous dynamical system, where trajectories can be obtained via numerical integration,

$$\dot{\mathbf{x}}(t) = g(\mathbf{x}(t)), \qquad \mathbf{x}(t) = \mathbf{x}_0 + \int_0^t g(\mathbf{x}(s))ds, \tag{4}$$

where $\mathbf{x}(t)$ give the task-space coordinates of the motion, $g : \mathbb{R}^3 \to \mathbb{R}^3$ is the dynamics, $\mathbf{x}_0$ is the initial condition for the trajectory, integrated for time $t$. For robot manipulators, where the Jacobian of the robot forward kinematics is known, we can further pull the dynamical system into the robot's configuration space by multiplying with the pseudoinverse of the Jacobian, as is done in [8].

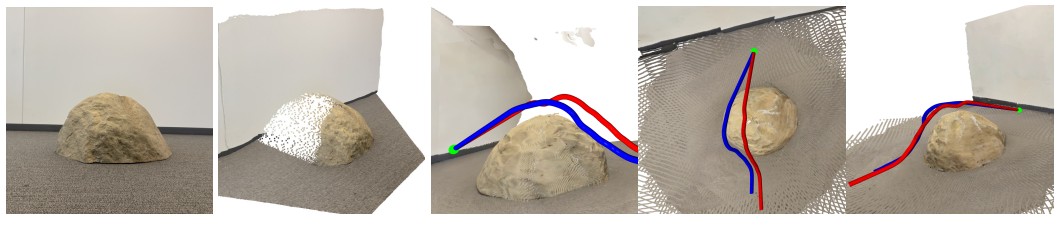

| (a) input | (b) DA-V2 | (c) Our reconstructed 3D representation and trajectories |

Figure 5: With a single example input image, shown in (a), of a stone model in an indoor environment, VGER can build a 3D representation of the scene. It does not suffer from incomplete surfaces like the results from DepthAnything-V2, shown in (b)). It can also facilitate downstream motion trajectory generation. Two trajectories, colored in red and blue, smoothly avoiding the obstacle, are shown in (c).

**Environment Influence as a Metric Field:** We propose a principled framework for shaping motion generation based on task-driven Riemannian metrics derived from the upstream distance function $f_\theta$, such that the reactive behaviours like collision-avoidance and surface following emerge. Given an nominal base dynamical system $g_{base}(\mathbf{x})$, we define a metric-modulated motion as

$$\dot{\mathbf{x}} = G(\mathbf{x})^{-1} g_{base}(\mathbf{x}), \qquad \text{and} \quad G(\mathbf{x}) \in \mathbb{S}^{++}. \qquad (5)$$

Here, $g_{base}$ is the dynamics that governs the initial motion pattern without considering the environment. This could be defined as a simple attractor to a goal coordinate or a dynamical system learned from data [33, 34]. $G(\mathbf{x})$ is a Riemannian metric that is positive definite ($\mathbb{S}^{++}$) and varies smoothly throughout the environment, constructed from $f_\theta$. This metric-based modulation can be interpreted as solving, at every point $\mathbf{x}$, a local quadratic program,

$$\dot{\mathbf{x}} = \arg\min_{\mathbf{v} \in \mathbb{R}^3} \left\{ \frac{1}{2} \mathbf{v}^\top G(\mathbf{x}) \mathbf{v} - \mathbf{v}^\top g_{base}(\mathbf{x}) \right\}. \qquad (6)$$

Thus, the metric shaping procedure can be viewed equivalently as solving a structured QP at every point in state space, where the metric $G(\mathbf{x})$ defines the local cost geometry. Importantly, by modulating $G(\mathbf{x})$, we influence both the preferred directions of motion and the associated effort, enabling smooth reactive collision avoidance.

Our goal is now to construct a Riemannian metric field to induce smooth collision avoidance behavior around surfaces represented by a learned unsigned distance function $f_\theta : \mathbb{R}^3 \to \mathbb{R}_{\geq 0}$, where $f_\theta(\mathbf{x})$ approximates the distance from a point $x \in \mathbb{R}^3$ to the closest point on the surface. To penalize motion toward the surface, we construct the collision avoidance metric as

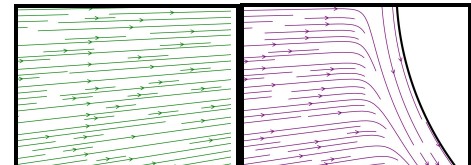

$$G(\mathbf{x}) = I + f_{\text{blow}}(\mathbf{x}) \, \mathbf{u}(\mathbf{x}) \mathbf{u}(\mathbf{x})^\top, \qquad (7)$$

where $I$ is the $3 \times 3$ identity matrix. Here, define a blow-up scaling factor $f_{blow}$ and denote a unit surface normal $u(\mathbf{x})$, where,

Figure 6: Constructed metric field stretches and warps a base dynamical system $g_{base}(\mathbf{x})$ with flows (left), to produce the flows $G(\mathbf{x})^{-1} g_{base}(\mathbf{x})$ which avoid colliding into the black surface (right).

$$f_{blow}(\mathbf{x}) = \frac{k}{(f_\theta(\mathbf{x}) + \epsilon)^4} \exp(-\beta f_\theta(\mathbf{x})), \qquad \mathbf{u}(\mathbf{x}) = \frac{\nabla f_\theta(\mathbf{x})}{\|\nabla f_\theta(\mathbf{x})\|}. \qquad (8)$$

Here, $k > 0$ controls the penalty strength, $\epsilon > 0$ ensures numerical stability near the surface, and $\beta > 0$ induces exponential decay away from the surface. Direct inversion of the metric $G$ at every control step would incur a full $3 \times 3$ matrix solve. Instead, we exploit the fact that $G$ is a rank-one update of the identity and apply the Sherman–Morrison formula [35] to eq. (5):

$$\left(I + \alpha \, \mathbf{u} \, \mathbf{u}^\top\right)^{-1} = I - \frac{\alpha}{1+\alpha} \, \mathbf{u} \, \mathbf{u}^\top \iff \dot{\mathbf{x}} = \left[I - \frac{f_{\text{blow}}(\mathbf{x})}{1 + f_{\text{blow}}(\mathbf{x})} \mathbf{u}(\mathbf{x}) \mathbf{u}(\mathbf{x})^\top\right] g_{\text{base}}(\mathbf{x}). \qquad (9)$$

As the trajectory approaches a surface, that is $f_\theta(\mathbf{x}) \to 0$, the metric sharply penalizes motion aligned with the surface normal, naturally steering trajectories tangentially to the surface without

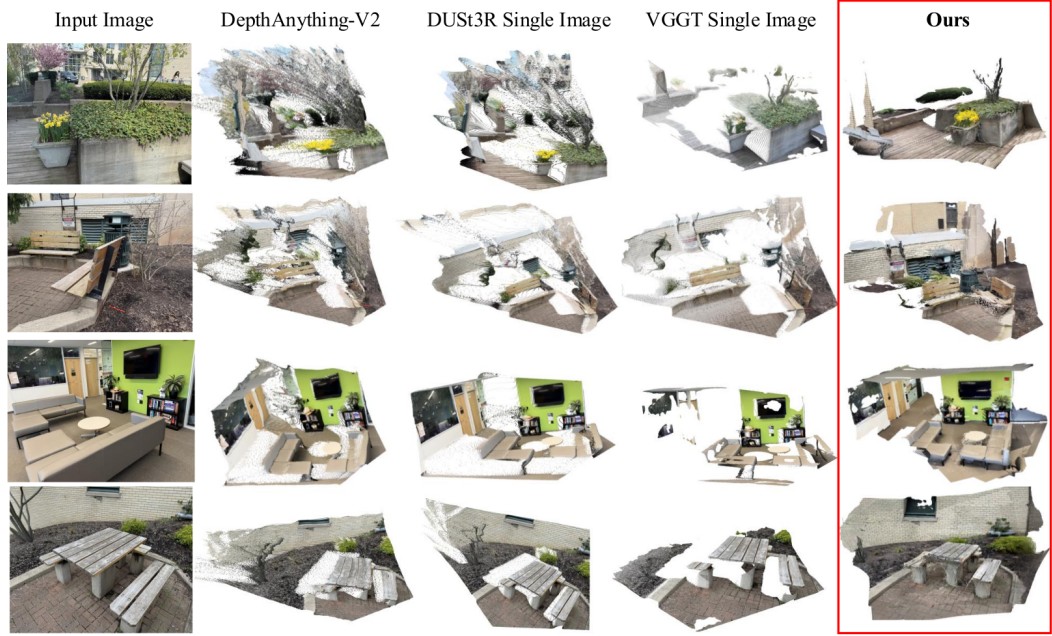

| Input Image | DepthAnything-V2 | DUSt3R Single Image | VGGT Single Image | **Ours** |

Figure 7: Qualitative evaluations of the complex 3D environment scenes (from top to bottom: garden, bench, indoor, table respectively) each extracted from a single image with our VGER relative to the baseline, DepthAnything-V2 and DUSt3R/ VGGT models with a single image input. We observe that not only does VGER fill in gaps occluded in the single image, but also altogether avoids the frustum-shaped noise artifacts, which are particularly prominent in the garden and bench images.

requiring hard constraints. The construction is smooth and fully differentiable, making it well-suited for reactive online motion generation. We illustrate the effect of a metric field steering the vector field away from the normal of the surface in fig. 6, on a toy problem. Here, we observe that the illustrated base vector field (shown in the left), after applying the effect of the metric field, smoothly warps around the black surface.

## 4 Empirical Evaluations

To evaluate the robustness and quality of our proposed Video-generation Environment Representation (VGER) along with the produced motion trajectories, we collect sets of 10 images in both outdoor and indoor scenes. These include the complex multi-object scenes **Garden**, **Bench**, **Table**, and **Indoor**, along with the more object-centric **Stone** and **Cabinet** scenes. For the evaluation of the quality of the performance of VGER and benchmark models, we extract the structure from a single image. We use the entire set of ten images, passed to DUSt3R, to construct a representation that we then consider to be the ground truth. Implementation details in our experiments are provided in section A1.

### 4.1 VGER Avoids Frustum Noise and Extracts Structure Accurately

We compare VGER with a single input image against a suite of strong baselines. In our setup, we uniformly sample 10 frames from the generated video to extract out our structure for VGER. These include: **DepthAnything-v2** [9]: a state-of-the-art monocular depth estimation foundation model; **VGGT with a single image** [15]: a recent 3D foundation model which can handle a single input image. We only pass a single image to generate the 3D representation; **DUSt3R with a single image** [6]: a widely used 3D foundation model, we again only input a single image to generate the 3D representation; **GEN3C** [20] and **ViewCrafter** [21]: These are contemporary controllable video generators, which we then extract 3D structure from to obtain the representation.

To evaluate the quality of the extracted scene representation, we rescale the environments to the interval $[-1, 1]$ and align the extracted scene representations with the ground truth representation built with the entire set of images using iterative closest point [36]. Then, we compute the Chamfer

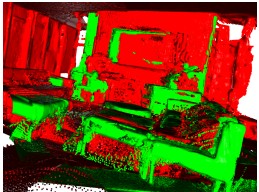

| | Garden | Bench | Table | Office | Stone | Cabinet |
|---|---|---|---|---|---|---|
| DepthAnythingV2 | 0.222 | 0.033 | 0.065 | 0.065 | 0.259 | 0.148 |
| VGGT (single image) | 0.109 | 0.068 | 0.094 | 0.062 | 0.296 | 0.121 |
| DUSt3R (single image) | 0.266 | 0.049 | 0.045 | 0.054 | 0.283 | 0.090 |
| GEN3C [20] | **0.060** | **0.016** | 0.060 | 0.049 | 0.104 | 0.066 |
| ViewCrafter [21] | 0.086 | 0.028 | 0.066 | 0.057 | 0.178 | 0.075 |
| VGER (Ours) | 0.075 | 0.029 | **0.030** | **0.047** | **0.096** | **0.044** |

Figure 8: (Left) Before computing distances, structure from one image by VGER (green) vs. ground-truth (red) are aligned. (Right) We report normalized Chamfer distances on different scene categories. Lower is better.

Distance [37] between the point clouds of the ground truth and extracted structure. For two point clouds $P_1$ and $P_2$, this is defined as,

$$D_{\text{Chamfer}}(P_1, P_2) = \frac{1}{|P_1|} \sum_{x \in P_1} \min_{y \in P_2} \|x - y\|_2 + \frac{1}{|P_2|} \sum_{y \in P_2} \min_{x \in P_1} \|y - x\|_2. \qquad (10)$$

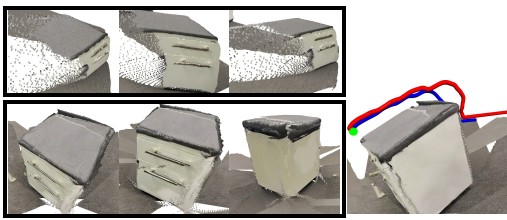

Figure 9: Single-image extraction leaves occluded regions incomplete, VGER reliably reconstructs them. Left: Top: DepthAnything-V2; bottom: VGER.; Right: Trajectories approaching the cabinet from the occluded backside, avoiding collision to reach the green goal.

The qualitative results are presented in fig. 8. Additionally, we illustrate an example alignment of the structure of the indoor environment extracted from a single image with VGER overlaid with a structure extracted via a 3D foundation model using all of the images collected. The Chamfer distances are subsequently computed between the aligned structures to obtain our quantitative results. We observe that VGER outperforms all the baseline methods consistently. Qualitatively, we observed that by using video generators, conditional on the input image, VGER subjects our structures to greater multi-view consistency. As a result, the frustum-shaped noise artifacts that appear in free space no longer appear in our obtained structure. Additionally, the video generator successfully estimates much of the geometry occluded from a single image, resulting in more geometrically accurate representations. This is particularly prominent in the complex Bench and Table scenes shown in rows 2 and 4 in fig. 7.

Here, we also highlight VGER's ability to deal with obstructions in fig. 9, where VGER can reconstruct the occluded backside of a cabinet. This enables motion trajectories and the correct collision-avoidance behaviour of even when approaching the cabinet from the back, which was not observed from the input image. A more intricate level of detail in regions that are hidden from the input image can also be seen in fig. 10. Here, by leveraging the large amounts of video data used to train the video generator model,

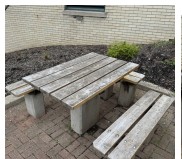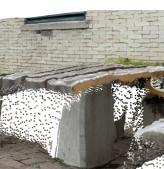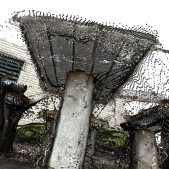

Figure 10: Left: Input image; Middle: Comparisons do not reconstruct unseen areas; Right: VGER reconstructs geometrically-consistent details in unseen regions, such as under the table.

VGER is able to accurately capture the geometry of the reverse side of the table surface, as well as the structure under the table, hidden from the input image view. We observe that this cannot be captured by the alternative comparison methods, and are left with gaps in obstructed regions.

## 4.2 Trajectories from Motion Policies Built on Constructed Representations

Next, we assess how VGER improves downstream motion generation. Using the approach outlined in section 3.3, we generate trajectories with implicit distance models derived from VGER, DepthAnything-V2 (DA-V2), and our ground-truth geometry. Our metric-field-based motion policy yields smooth trajectories, exemplified in the table scene (fig. 11), and we can compute per-step velocities in under 1ms. To quantify alignment with true motion, we integrate collision-free paths

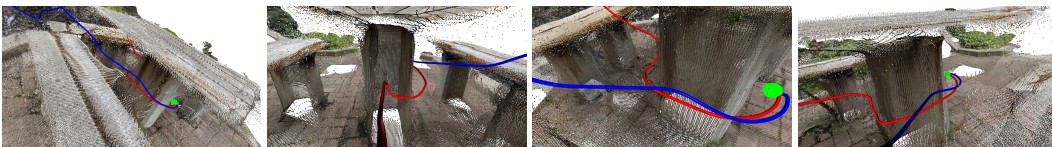

Figure 11: VGER enables smooth collision-free motion. Here, two trajectories (red and blue) warp over benches and around the leg of the table to reach the goal (green).

| 3 pics | 5 pics | 7 pics | 10 pics | 15 pics |
| --- | --- | --- | --- | --- |

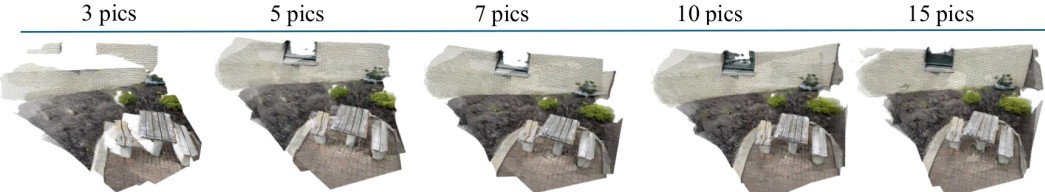

Figure 12: The extracted structure provided by VGER, sampling different numbers of frames from the generated video.

to a designated goal and measure the Discrete Fréchet Distance (DFD) [38], normalized by the Euclidean distance from start to goal. As shown in fig. 13c, lower DFD values indicate closer agreement with the ground-truth trajectories. Across all test scenes, VGER-based structures consistently outperform DA-V2, with particularly large gains in indoor, cabinet, and stone environments. This advantage arises because alternative methods introduce frustum-shaped noise artifacts in free space, which "funnel" the motion into occluded regions (see fig. 13a). By contrast, VGER completes obstructed areas without spurious artifacts, guiding trajectories correctly around the backside of the cabinet. Further examples in the complex indoor and garden scenes (fig. 13b) demonstrate that VGER-driven policies smoothly avoid collisions with the table surface and elegantly skirt around the plant and curb, respectively.

### 4.3 Ablation: Frames of Generated Video Used for Structure Extraction

The key to VGER's performance is its ability to leverage frames of videos generated from a large pre-trained model. When extracting the structure from the video, it can be unnecessarily resource-intensive to use every frame in the video, and we can uniformly subsample frames instead. In our results reported, we subsample 10 images.

Table 1: Performance of VGER with various images sampled on the *table* scene.

| # Images | 3 | 5 | 7 | 10 | 15 |
| --- | --- | --- | --- | --- | --- |
| Dist. ($\times 10^{-2}$) | 4.2 | 3.1 | 3.3 | 3.0 | 2.9 |

Here, we conduct an ablation study over the number images sampled from the video. We report results on the *table* scene using 3, 5, 7, 10, 15 images, computing the normalized Chamfer distance between the extracted structure and the ground truth after alignment. We observe that, beyond 5 images, the performance of VGER is robust to the number of frames extracted from the generated video. Images of the structures extracted for number of images sampled are in fig. 12.

## 5 Limitations

The VGER pipeline exploits pre-trained video generators as foundation models for reconstruction and motion generation. In practice, these models synthesize geometry well within the camera's field of view but may "hallucinate" when extrapolating beyond the observed frustum. As the spatial distance from the original image capture region increases, so too does the uncertainty. An example of hallucinations in the generated video is illustrated in fig. 14. Future avenues of research

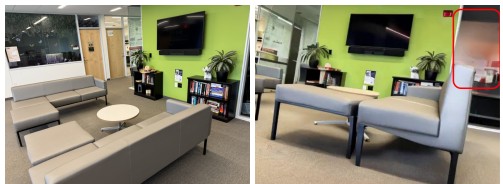

Figure 14: Left: The input image; Right: Generated frame contains hallucinated structure outlined in red that does not match the setting.

to extend VGER can incorporate mechanisms to track extrapolated regions far outside the spatial re-

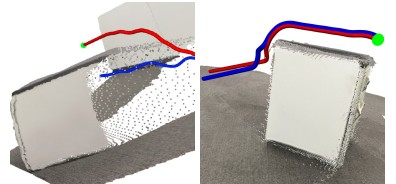 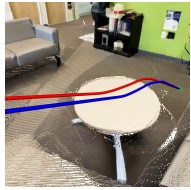 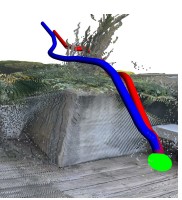

| | Ours | DA-V2 |
|---|---|---|
| Bench | 0.087 | 0.098 |
| Garden | 0.232 | 0.245 |
| Indoor | 0.111 | 0.203 |
| Table | 0.211 | 0.245 |
| Cabinet | 0.127 | 0.322 |
| Stone | 0.109 | 0.190 |

(a) Cabinet, Left: DA-V2; Right: VGER.      (b) VGER trajectory examples.      (c) Normalized DFD.

Figure 13: (a) Frustum-shaped noise artifacts often introduce local minima in the motion objective. We compare trajectories generated using DepthAnything-V2 (DA-V2) versus VGER: because VGER completes the cabinet's backside without spurious artifacts, it successfully navigates to the goal. (b) Sample paths on VGER-derived structures smoothly traverse along the table surface (indoor) and curve around a plant (garden), reaching their targets without collision. (c) The normalized Discrete Fréchet distances between trajectories generated with VGER and DA-V2 structures.

gion captured by the input image, assign uncertainty estimates and integrate the confidence-levels within the downstream motion policies.

# 6   Conclusion

In this paper, we tackle the problem of building an environment representation that enables motion generation from a single image. Here, we have present Video-generation Environment Representation (VGER), a framework that leverages pre-trained video synthesis and 3D foundation models to recover dense scene geometry from a single RGB image, without frustum-shaped artifacts. By applying a multi-scale noise contrastive sampling procedure to the 3D reconstruction, we extract an implicit unsigned distance field that smoothly encodes obstacle proximity across the workspace. Embedding this field into a metric-modulated motion framework yields a motion policy, modelled by a non-linear dynamical system capable of producing collision-free trajectories directly from a single viewpoint. Extensive evaluations demonstrate that VGER consistently outperforms state-of-the-art baselines, avoiding frustum-shaped artifacts and smoothly navigating complex indoor and outdoor scenes. This drives the broader impact of a future where robots can be actively deployed in open environments, enabling a safer workplace in the future.

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

# Appendix

## A1  Implementation Details

We run our experiments on a standard desktop with an Intel i9 CPU and an NVIDIA RTX 4090 GPU with 24GB VRAM. We use all the standard hyper-parameters in the Seva and DUSt3R foundation models used in our pipeline. Here list all the hyper-parameters used for the reconstruction and motion policies construction of our VGER method in table A1.

| Implicit Model Multi-Scale Sampling | | Network for Implicit Model | | Motion Policy Blow-up | |
|---|---|---|---|---|---|
| $\alpha_{\mathrm{surf}}$ | 0.5 | Layers | 3 | $k$ | 20 |
| $\alpha_{\mathrm{eik}}$ | 0.1 | Width | 256 | $\beta$ | 100 |
| $\sigma_{\min}$ | 0.0025 | $\omega_0$ | 25 | $\epsilon$ | $10^{-8}$ |
| $\sigma_{\max}$ | 0.1 | Training Epochs per LR | 2000 | | |
| $|\mathcal{P}|$ | 10000 | Learning rates (LRs) | 3e-4, 1e-4, 5e-5, 1e-5 | | |
| $|\mathcal{B}_s|$ | 5000 | | | | |

Table A1: VGER hyperparameters at a glance.

## A2  Additional Visualizations of Reconstructions

Here we provide additional visualizations of the structures extracted from VGER along with baselines. These are provided in fig. A1. Additionally, here we provide visualizations for our ablation

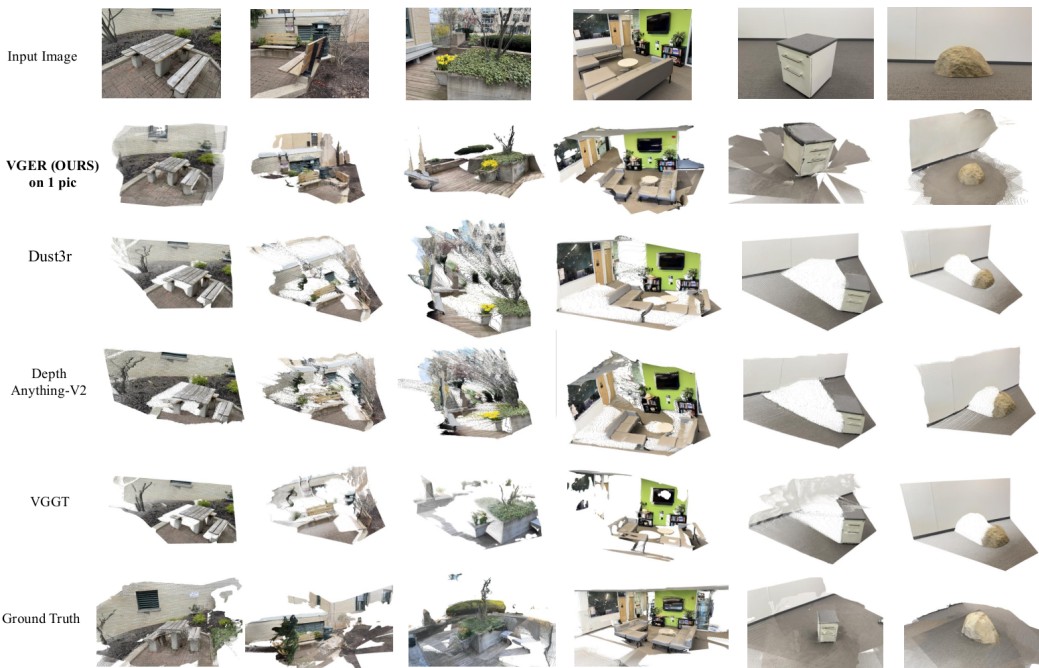

Figure A1: The extracted structures from our method, baselines, and ground truth, along with the single input image, for the table, bench, garden, indoor, cabinet, and stone environments.

experiment, and observe the structure extracted via VGER when the number of frames that we use from the generated video differ.

