# OpenReview forum: "Building 3D Representations and Generating Motions From a Single Image via Video-Generation"
_NeurIPS.cc/2025/Conference — NeurIPS 2025 poster_

### Official Review · Reviewer_NSXd · 2025-06-02

**Clarity:** 3
**Significance:** 3
**Originality:** 2
**Rating:** 4
**Confidence:** 3

**Summary:**

This paper presents a pipeline for generating robust collision-free motion policies from single images. Their framework consists of three parts: lifting an image into a complete 3D scene, training an implicit unsigned distance field (UDF) representation of the scene, and incorporating the implicit metric field into the motion planning for collision avoidance. To lift an image into a 3D scene, they observe that the simplest implementation of unprojected depth map suffers from two artifacts — noisy depth points along surface boundaries and incomplete surfaces — and use a camera-conditioned video generation method to generate unseen views and a feedforward SfM method (dust3r) to lift it into a 3D point cloud. They then train an implicit field on the resulting 3D point cloud and incorporate the UDF into the motion planning for collision-avoidance.

**Questions:**

- Is there a simpler setup you can use? I would like to see more ablations in-between unprojected depth estimate and your full model (mentioned more in-depth in weaknesses section)
- Why not just use resulting DUST3R point cloud in motion planning? (mentioned above)
- How long does training the per-scene MLP take? Those details should be mentioned

**Ethical Concerns:**

["NO or VERY MINOR ethics concerns only"]

**Final Justification:**

I think this paper is definitely still in the borderline accept region and not a 'definite accept', but I also don’t have much expertise in motion planning (more so the 3d vision side), so I am not 100% sure.
The idea is interesting but the implementation is a bit impractical in my estimation (waiting several minutes for optimization rather before the robot can make a motion plan / act). The evaluations are also a bit weak in terms of simpler evaluations and baselines and no real-world demonstration. I think if it is accepted, they have to definitely address the speed of optimization in the paper as limitation and future work (or at the very least include the optimization times in the paper).

**Limitations:**

- Method requires per-scene training of an MLP before motion plan can be generated, which is impractical for most robotics settings
- Hallucinated content is used with no uncertainty measuring (one could e.g. render multiple videos and check variance or generated videos as proxy for uncertainty)

**Paper Formatting Concerns:**

No major issues.

**Quality:**

3

**Strengths And Weaknesses:**

Strengths:

- Good direction of using 3D foundation models in robotics motion planning with minimal/no training overhead
- Paper writing is clear and easy to follow

Weaknesses:

- Engineering focused — while I agree the framework is useful, I do think it is a bit plug-and-play and it’s worth noting there perhaps isn’t much technical novelty
- Clunky and expensive — the framework has a few distinct moving pieces without interaction between them: the video generation model, feedforward SfM model, per-scene UDF model. Perhaps most clunky is that the UDF needs to be trained per scene. Either inferring this feedforward or avoiding with some heuristic geometry could simplify the pipeline and make it more practical
- Experiments and ablations and pipeline could be improved — currently the authors compare their motion plans vs those from the unprojected depth anything results. I personally would be interested in the following ablations:
    - what about heuristic geometries like meshes or using the raw point cloud from DUST3R
    - heuristically cleaning the pointcloud with a depth-gradient thresholded mask like e.g. MoGe does in their visualizations
    - using the pointcloud from an image-to-pointmap representation like single-frame VGG/DUST3R
    - using some unseen-geometry heuristic like don’t cross any part of space unobserved (implemented as some geometry proxy if feasible)

---

> ### Author Rebuttal · Authors · 2025-07-31
>
> We thank the reviewer for their constructive and extremely helpful feedback. We appreciate the positive feedback of our methodology and the clarity and presentation of the manuscript.
>
> Here, we’d like to answer some of the questions presented. Our proposed VGER goes directly from a single RGB image to a scene, and also attempts to provide estimates in regions that are occluded. Generally, the reconstructed surfaces provided by 3D foundation models are of higher quality (less noise) than off-the-shelf RealSense cameras. With a good stationary depth image, I would expect it not to have the frustum-shaped errors that are common when trying to extract structure from monocular depth estimation models (e.g. DepthAnything). However, VGER still provides additional value by enabling the generation of regions that are occluded.
>
> We agree that the resulting point cloud from the 3D foundation models could perhaps be used for collision checking in motion planners such as RRTs. The construction of the implicit model enables a smoothly varying distance to the surface to be learned, enabling the construction of the policy, given as a vector field in 3D space. This will enable trajectories to be efficiently integrated at different initial positions, whereas sampling-based motion planners, which do collision-checking based on the point cloud, generally return a single trajectory/path, rather than a policy. We acknowledge that training the implicit model from the point cloud representation needs to be done offline. However, this is not overbearingly slow, and the per-scene training time is approximately several minutes on consumer-grade GPUs.

---

> > ### Comment · Reviewer_NSXd · 2025-08-04
> >
> > I see the author’s point that using the video model for unseen regions over just a depth estimate provides information about unseen regions, but I still think the per-scene optimization of the distance field is burdensome — training an implicit model that takes several minutes is likely prohibitive. I think therefore I would have liked to see the simplest ablation on using the raw point-cloud from foundation models after the video generation, rather than doing the distance field fitting. I also agree with reviewer pvyM on the weakness of not having any real-world evaluation.
> > At the very least I think the training time needs to be posted in the paper and listed as a serious limitation and the authors should provide some roadmap towards reducing that optimization time, such as faster optimizers, early stopping, or just using foundation model geometry without optimization altogether.
> > In general I think the premise of using the video generation model for the geometry of unseen parts of the scene in the context of motion planning is promising, but the exact implementation is still a bit burdensome.

---

### Official Review · Reviewer_pvyM · 2025-06-22

**Clarity:** 3
**Significance:** 3
**Originality:** 4
**Rating:** 4
**Confidence:** 3

**Summary:**

This paper proposes VGER, a novel pipeline that generates environment-aware motion policies from a single RGB image. The key idea is to leverage large video generation models (e.g., Seva) to synthesize a multiview video from the input image, and then reconstruct a dense 3D scene using a 3D foundation model (e.g., DUSt3R). An implicit distance field is trained using multi-scale noise contrastive supervision, and motion policies are then derived using a Riemannian metric field constructed from this geometry. The method is evaluated on diverse indoor/outdoor scenes, demonstrating improved reconstruction quality and smoother, collision-free motion trajectories compared to monocular depth baselines.

**Questions:**

How robust is VGER to real-world image conditions (e.g., noise, lens distortion)? Could you provide an ablation or example on natural images? Have you considered learning the Riemannian metric instead of hardcoding it? This could help generalize better in cluttered or ambiguous scenes. Could confidence maps from the 3D foundation model be propagated into the motion planning stage to avoid unreliable areas? Do you plan to test this on a real robot or photorealistic simulator?

**Ethical Concerns:**

["NO or VERY MINOR ethics concerns only"]

**Final Justification:**

The authors' responses address key concerns effectively—real-world images confirm applicability beyond synthetics, and commitments to uncertainty handling, metric learning, and robot testing enhance potential impact. While ablations on noisy conditions and larger/dynamic scenes are still missing, the core innovation and results hold up well. Reasons to accept (elegant integration, strong visuals, robotics relevance) outweigh remaining limitations (e.g., hallucination handling, evaluation scope). I therefore keep my score for borderline accept

**Limitations:**

yes

**Quality:**

3

**Strengths And Weaknesses:**

**Strengths**: (1) The integration of pretrained video generators with 3D foundation models to create multiview depth from a single image is an innovative and elegant approach. (2) The modular design (video synthesis → point cloud → implicit field → motion policy) is clean and extensible. (2) The qualitative results is compelling. Visuals across several scenes show smoother, safer motion paths and less noisy geometry compared to strong baselines like DepthAnything-V2. (4) The construction of the implicit field via multi-scale noise denoising is well-grounded and connected to score-based modeling.

**Weaknesses**: The authors acknowledge hallucinations from video generation but provide no mechanism to handle or flag unreliable geometry during motion planning. Also, no real-world to sim-to-real validation is conducted, all experiments are in synthetic environments. Demonstration on it would work on actual robots or noisy RGB images can be helpful. The Riemannian field used to guide motion is hand-designed. There’s no ablation or learning-based alternative explored. The benchmark scenes are mostly static and small-scale. The method is not evaluated in dynamic or large outdoor settings.

---

> ### Author Rebuttal · Authors · 2025-07-31
>
> We thank the reviewer for their constructive comments and for highlighting the performance and innovation of our approach!
>
> We wish to highlight that the pictures of both the indoor and outdoor scenes were images that we took in the real-world, so the performance seen is indicative of performance on real images. We agree with the suggestions here that the uncertainty from the 3D foundation model can be propagated to the VGER model to guide the motion generation, and potentially achieve active perception by enabling robots to actively perceive these regions. This shall be explored in subsequent work. Additionally, the Riemannian Metric here is constructed such that collision avoidance is achieved. However, you’re completely right that this metric can also be learned to, for example, imitate a set of provided demonstrations. Finally, we definitely plan to deploy the method onto a real robot and have identified table-top scenes to be particularly suitable, as VGER would be able to estimate regions of the scene which are occluded, which would be particularly valuable.

---

> > ### Comment · Reviewer_pvyM · 2025-08-04
> >
> > Thank you to the authors for the rebuttal and clarifications. I appreciate the confirmation that the evaluation uses real-world images, which partially addresses my concern about sim-to-real validation—though a brief real-robot demo or noisy image ablation would still strengthen the work. The planned table-top robot deployment for occluded regions sounds particularly relevant. Overall, this bolsters the practical impact, but the limited evaluation on dynamic/large scenes remains a gap.
> > Reasons to accept (elegant integration, strong visuals, robotics relevance) outweigh remaining limitations (e.g., hallucination handling, evaluation scope). I will maintain my positive rating.

---

### Official Review · Reviewer_xqDw · 2025-07-01

**Clarity:** 3
**Significance:** 2
**Originality:** 1
**Rating:** 4
**Confidence:** 4

**Summary:**

This paper focuses on the problem of collision-free motion generation. It leverages large-scale video generation models to generate multiple views for the static scene, which is used to construct the 3D dense point cloud through 3D foundation models. Then, it generates collision motion with motion models.

**Questions:**

* I'm not familiar with the related works about motion generation. Is it the first work to generate motion based on a 3D dense point cloud?
* What's the efficiency of the multi-scale noise motion generation?
* What's the hyperparameter of the motion generation models? What's the number of the point? If it's a sparse point cloud, how does the motion generation model perform?

**Ethical Concerns:**

["NO or VERY MINOR ethics concerns only"]

**Final Justification:**

1. The method demonstrates strong performance in scene construction.
2. The motion policies seem robust and novel, as claimed by the authors.
3. The baseline comparisons are relatively simplistic.

**Limitations:**

yes

**Quality:**

2

**Strengths And Weaknesses:**

* Strength
* The method demonstrates strong performance in scene construction. The approach achieves impressive results in building dense 3D point clouds by combining video generation models with 3D reconstruction models.
* The paper is well-written and good presentation, making the methodology and results accessible and understandable.
* Weakness
* The proposed 3D construction approach is closely similar to existing work, particularly ViewCrafter [1], which similarly integrates video diffusion models with 3D reconstruction for novel view synthesis. This similarity raises concerns about the novelty of the contribution.
[1] ViewCrafter: Taming Video Diffusion Models for High-fidelity Novel View Synthesis.
* The baseline comparisons are relatively simplistic. The paper mainly contrasts against 3D reconstruction methods such as Dust3R and VGGT applied to single images, which may not reflect the improvement of the methods.
* And there is no comparison of motion generation methods with other works.
* The actual impact of motion generation on downstream robotics tasks is not thoroughly demonstrated. The paper lacks a clear explanation or empirical evidence on how the generated motions contribute to improving performance or capabilities in practical robotic applications.

---

> ### Author Rebuttal · Authors · 2025-07-31
>
> We thank the reviewer for the thoughtful comments and for highlighting both the performance and presentation of the manuscript.
>
> We thank the reviewer for pointing out additional baselines for experimental comparisons. Here we report normalized Chamfer distances of VGER as compared to the contemporary methods, ViewCrafter and GEN3C:
>
> | Method          | Tables and Benches | Indoor Office | Stone | Garden | Bench | cabinet |
> | --------------- | ------------------ | ------------- | ----- | ------ | ----- | ------- |
> | **VGER**        | **0.030**              | **0.047**         | **0.096** | 0.075  | 0.029 | **0.044**   |
> | **ViewCrafter** | 0.066              | 0.057         | 0.178 | 0.086  | 0.028 | 0.075   |
> | **Gen3C**       | 0.060              | 0.049         | 0.104 | **0.060**  | **0.016** | 0.066   |
>
> The comparison results are tabulated above. We observe that all three of the methods have solid performance, as measured by chamfer distance. VGER generally outperforms the alternative methods, with GEN3C also having strong performance in the garden and bench scenes. Additionally, VGER and ViewCrafter have comparable VRAM and runtime costs while GEN3C requires more VRAM and time.
>
> | GPU             | VRAM  | Time      |
> | --------------- | ------- | --------- |
> | **GEN3C**       | 42.9 GB | \~30 mins |
> | **ViewCrafter** | 22.6 GB | \~4 mins  |
> | **VGER**        | 19.7 GB | \~7 mins  |
>
> The introduced method is the first reactive approach, which produces a policy (velocity field) in the 3D environment that leverages an implicit representation constructed from a dense point cloud. Alternative approaches such as motion planning via sampling-based motion planners and trajectory optimization approaches generally produce a single path or trajectory rather than a vector field which can be integrated. In that sense, the proposed approach is reactive, and is conceptually similar to Riemannian Motion Policies (RMPs) (Ratliff 2018) and Geometric Fabrics (Van Wyk, 2024). These approaches typically assume that the environment is constructed of a set of primitive shapes. The training of the implicit method with the multi-scale noise sampling is fairly efficient and can be achieved in minutes on consumer-grade GPUs.
>
> The hyperparameters for the motion policies constructed are in Table A1 in the appendix under “Motion Policy Blow-up”, we will make sure to make this more prominent in the next version of this manuscript. Additionally, the number of points in the representation before implicit representation is constructed differs from scene to scene, depending on the confidence outputted by the 3D foundation model, but is generally in the millions of points. We also thank the reviewer for emphasizing the connection with robotics tasks. In this paper, we focused on relatively larger both indoor and outdoor scenes. An interesting scene that further highlights the benefits of our approach is table-top scenes to do motion generation for manipulators. This is an area where predicting complete geometries can prove to be very important, we will add this scene into the next version of our submission.

---

> > ### Comment · Reviewer_xqDw · 2025-08-05
> >
> > Thanks for the response.
> >
> > I'm curious about how motion policies perform when the point cloud is sparse or imperfect.
> > Are complete and perfect geometries needed for motion policies? If so, I think this is a large limitation.
> >
> > Additionally, I also wonder why VGER is better than ViewerCrafter? Is it because the video generation model or 3D foundation model?

---

> > > ### Author Response · Authors · 2025-08-06
> > >
> > > Thanks for the reply! The motion policies are generally robust under noise and sparse point clouds, as the training of the implicit model results in a slightly smoothed and interpolated surface. Generally, the video generation will fill in gaps that are occluded, such as under the table (shown in fig 13) or the backside of the drawer (shown in fig 9.), which were not in the input image.
> > >
> > > ViewCrafter contains traits which are similar to our VGER. It leverages estimates a dense point cloud, then leverages a video generator conditioned on the partial point cloud to fill in the missing regions, enabling novel view generation. VGER operates in a reverse order by conditioning on the image to generate a video sequence, then constructs the geometry from the crafted video. Although we don't have a definitive response for why VGER outperforms. From our observations, a key hypothesis for why the performance of VGER seems to edge out is that image-conditioned video generation models may produce higher-quality videos. On top of that, ViewCrafter generally focuses the camera trajectories to fill in enclosed missing regions, while VGER can extend beyond the edges of the original input image.

---

### Official Review · Reviewer_V16n · 2025-07-02

**Clarity:** 3
**Significance:** 2
**Originality:** 2
**Rating:** 5
**Confidence:** 3

**Summary:**

This paper leverages the additional information provided by a video generation model to perform 3D reconstruction from a single image using Dust3R, and subsequently builds an implicit representation to enable a collision-free motion generation algorithm. Specifically, after generating a video to obtain multi-view images, it merges the point clouds estimated by Dust3R (filtered by confidence) across views to reconstruct the 3D scene. It then trains an implicit distance field with multi-scale noise, and finally generates a motion plan based on this representation.

**Questions:**

1. **Implicit scene description**
   Are there prior works using implicit scene representations for collision avoidance? How does this method compare in terms of accuracy or efficiency?
2. **Failure cases / initial image impact**
   Can you show more failure cases or analyze how the initial image content affects reconstruction and motion planning?
3. **Reliability**
   Are there ways to improve reliability — for example, by incorporating active perception, sensor fusion, or uncertainty estimation?

**Ethical Concerns:**

["NO or VERY MINOR ethics concerns only"]

**Final Justification:**

Though the use case of robot path search seems a little weird to me given the time cost and VRAM cost, I still think the 3D reconstruction part of the method have enough contribution to be accepted, as it gets better reconstruction result with less time-cost comparing to prior works.

**Limitations:**

Yes.

**Quality:**

3

**Strengths And Weaknesses:**

**Strengths:**

1. The method effectively leverages a video generation model to fill in the missing information from single-image Dust3R point clouds, achieving impressive 3D reconstruction results.
2. This is the first work to address the task of generating robot collision-free trajectories from single images by integrating perception and motion planning processes.
3. The paper introduces a method to train an implicit distance model and continuous metric field that can efficiently generate motions based on the environment’s representation.

**Weaknesses:**

1. There are already many existing frameworks that use video generation (or multi-view diffusion) for 3D reconstruction [1, 2, 3], so the novelty of this contribution is questionable. At minimum, the paper should include a thorough literature review and provide a clear justification for why this approach offers advantages over prior work.
2. As noted in the limitations, video generation models fail to produce accurate representations of parts of the scene that are not visible in the input image. The success of the method thus heavily depends on the input image capturing most of the scene.
3. The reliability of the approach is uncertain. For robot motion planning, the required reliability is higher than for typical vision tasks. For example, as shown in Figure 5, even a human observer cannot confidently infer what lies behind the stone—there are many possible variations. Relying on the “imagination” of a video generation model may not provide sufficient safety. Active perception strategies might be necessary.
4. A final concern is the computational cost and speed of the method. Running foundation models can be time-consuming and resource-intensive. In many cases, using a simpler solution, such as a depth sensor, might be more practical.

[1] Gao, Ruiqi, et al. "Cat3d: Create anything in 3d with multi-view diffusion models." arXiv preprint arXiv:2405.10314 (2024).

[2] Yu, Wangbo, et al. "Viewcrafter: Taming video diffusion models for high-fidelity novel view synthesis." arXiv preprint arXiv:2409.02048 (2024).

[3] Ren, Xuanchi, et al. "Gen3c: 3d-informed world-consistent video generation with precise camera control." Proceedings of the Computer Vision and Pattern Recognition Conference. 2025.

---

> ### Author Rebuttal · Authors · 2025-07-31
>
> We thank the reviewer for their constructive comments and efforts in helping us improve the manuscript. Thank you for highlighting the methodological contributions and empirical results in our work.
>
> We particularly appreciate the reviewer taking the time to outline the methods, Cat3D, Viewcrafter, and Gen3C. These are contemporary methods which share similarities with VGER. In particular, Viewcrafter similarly leverages the 3D foundation model DUSt3R, Gen3C integrates a video generator, while VGER makes use of both.
>
> | Method          | Tables and Benches | Indoor Office | Stone | Garden | Bench | cabinet |
> | --------------- | ------------------ | ------------- | ----- | ------ | ----- | ------- |
> | **VGER**        | **0.030**              | **0.047**         | **0.096** | 0.075  | 0.029 | **0.044**   |
> | **ViewCrafter** | 0.066              | 0.057         | 0.178 | 0.086  | 0.028 | 0.075   |
> | **Gen3C**       | 0.060              | 0.049         | 0.104 | **0.060**  | **0.016** | 0.066   |
>
> The comparison results are tabulated above, we observe that all three of the methods have solid performance, as measured by chamfer distance. VGER generally outperforms the alternative methods, with Gen3C also having strong performance in the garden and bench scenes.
>
> | GPU             | VRAM  | Time      |
> | --------------- | ------- | --------- |
> | **GEN3C**       | 42.9 GB | \~30 mins |
> | **ViewCrafter** | 22.6 GB | \~4 mins  |
> | **VGER**        | 19.7 GB | \~7 mins  |
>
> The above table provides information of VRAM usage and runtime, we observe that VGER and ViewCrafter have relatively lower VRAM usage and runtime.
>
> Implicit representations, such as learned distance functions are typically used in trajectory optimization methods for motion generation. Our approach differs in that it is reactive, it does not require an entire trajectory to be found, and instead focuses on generating the immediate action. This enables reactive methods to be a lot faster and produces an entire policy (velocity field) rather than a single trajectory. This is conceptually the most similar to Riemannian Motion Policies (RMPs) (Ratliff, 2018), but unlike RMPs which require an environment representation constructed from primitive shapes, we allow for the learned implicit model to be used.
>
> We also appreciate the point raised about reliability, potential hallucinations in unseen regions,  and agree with the opportunity to leverage active perception. We note that we can actually track which regions of the reconstruction are connected to the original input image, and which regions are connected to frames generated via video generation. Additionally, the produced reconstruction from VGER includes pointwise confidence maps, which are inherited from DUSt3R. These confidence values can indeed be used to guide robots to actively perceive and reduce the uncertainties in areas not covered by previous images. However, we believe that producing an estimate of the environment structure even in unseen regions remains a valuable prior. Accomplishing object-level shape completion, for occluded objects, is valuable in robotics, likewise VGER accomplishes the completion of scene level occlusions not captured in a single image in one go. Overall, we definitely agree that active perception would be a practical solution to potential hallucinations described in the limitations section, and we hope to incorporate active perception as a follow-up to this work.

---

> > ### Comment · Reviewer_V16n · 2025-08-04
> >
> > My major concerns about the 3D reconstruction part are addressed by the new experiments, where the visual reconstruction method have an overall better result than previous works. This contribution outweighs my other concerns on the time cost and the VRAM cost (takes 7 minutes to plan a path). I will still lean to accept this paper.

---

> ### Comment · Reviewer_V16n · 2025-08-02
>
> What are the scores in table 1 mean?

---

> > ### Author Response · Authors · 2025-08-02
> >
> > The first table in the rebuttal contains normalized Chamfer distances on different scene categories, with additional methods for comparison, similar to the table in Fig. 8 in the manuscript.

---

### Note · Authors · 2025-08-16

We thank the reviewers for their constructive comments and suggestions in helping us improve our manuscript.

We appreciate the reviewers for highlighting the performance and innovation of our approach! Our method VGER introduces a pipeline that integrates the pretrained video generation models with 3D foundation models to produce multi-view videos and dense 3D reconstruction from a single RGB image. VGER demonstrates strong performance, outperforming contemporary methods such as diffusion-based methods (ViewCrafter, Gen3C), monocular depth estimators (DepthAnything-V2), and transformer-based models (DUSt3R, VGGT), while maintaining efficient VRAM and runtime costs.

In addition, VGER proposes the first reactive framework for policy generation in 3D environments. By exploiting an implicit representation built from a dense point cloud with a multi-scale noise-contrastive denoising approach, our method ensures that any nominal dynamical system coupled with the obstacle-induced curvature metric field will yield smooth, collision-free trajectories in real time.  Unlike sampling-based motion planners and trajectory optimization which output a single trajectory, our reactive approach produces an entire policy in less time. Compared to the conceptually similar approaches, Riemannian Motion Policies (RMPs) (Ratliff 2018) and Geometric Fabrics (Van Wyk, 2024), which require an environment representation constructed from primitive shapes, the training of the implicit method with the multi-scale noise sampling is fairly efficient and can be achieved in minutes on consumer-grade GPUs.

We appreciate the points raised by reviewers about reliability, potential hallucinations in unseen regions, and training time in per-scene optimization. We acknowledge that leveraging active perception and propagating the uncertainty from the 3D foundation model to the VGER model in guiding the motion generation. We will also note the training time as a limitation and outline avenues such as faster optimizers, early stopping, or hybrid methods to further reduce training time. Finally, we intend to deploy the methods onto a real robot in more scenes (table-top) in the next version of our submission.

In summary, VGER introduces the first framework that unifies video generation and 3D foundation models for motion policy learning directly from a single RGB image.  We believe these contributions advance the vision of robots deploying reliably and operating safely in open environments.

---

### Decision · Program_Chairs · 2025-09-17

**Decision:**

Accept (poster)

**Comment:**

After the rebuttal period, this paper received only positive reviews (3 borderline accept, 1 accept).

On the positive side, the reviewers consider the task (the combination of 3D reconstruction as well as collision-free motion policy) to be  interesting and the visual results compelling. On the more critical side, several of the reviewers questioned the technical novelty of the paper, noting that the 3D reconstruction community has explored the use of video generation quite extensively. While I tend to agree with this criticism, I overall agree with the reviewers that this line of work, particularly the motion policy side, seems promising and interesting.

I advocate for acceptance but would strongly encourage the authors to add a related work section discussing 3D reconstruction using video generators (e.g. the papers note by Reviewer V16n) as well as some rewriting.